# The Reshaping of the E-Cigarette Retail Environment: Its Evolution and Public Health Concerns

**DOI:** 10.3390/ijerph19148518

**Published:** 2022-07-12

**Authors:** Carla J. Berg, Albert Melena, Friedner D. Wittman, Tomas Robles, Lisa Henriksen

**Affiliations:** 1Department of Prevention and Community Health, Milken Institute School of Public Health, George Washington University, 950 New Hampshire Avenue, Washington, DC 20052, USA; 2George Washington Cancer Center, George Washington University, 800 22nd Street NW, #7000C, Washington, DC 20052, USA; 3San Fernando Valley Partnership, Inc., 1131 Celis Street, San Fernando, CA 91340, USA; amelena@sfvp.org (A.M.); trobles@sfvp.org (T.R.); 4CLEW Associates, 950 Gilman Street, Berkeley, CA 94710, USA; fwittman@clewassociates.com; 5Stanford Prevention Research Center, Department of Medicine, Stanford University School of Medicine, 3180 Porter Drive, Suite 125, Palo Alto, CA 94304, USA; lhenriksen@stanford.edu

**Keywords:** e-cigarettes, tobacco control, marketing, health disparities, social determinants

## Abstract

E-cigarette use represents a public health controversy in the US and globally. Despite the potential of e-cigarettes to support cigarette cessation, their use increases health risks and risk for addiction, particularly in young people. Various federal, state, and local laws have impacted tobacco retail in general and e-cigarettes in particular. In the US, 2019–2020 federal laws increased in the minimum legal sales age for tobacco to 21 and banned flavored cartridge-based e-cigarettes. Many states and localities were early adopters of Tobacco 21 and implemented more comprehensive flavor restrictions than the federal ban. Meanwhile, cannabis retail is increasingly being legalized in the US—while cannabis-based product regulation has notable gaps at the federal, state, and local levels. These regulatory complexities have impacted specialized retailers selling e-cigarettes, including “vape shops” that exclusively sell e-cigarettes, “smoke shops” that sell e-cigarettes and other tobacco (and potentially CBD/THC and other un- or under-regulated products), and online retail. This commentary outlines public health concerns related to: (1) youth access; (2) consumer exposure to a broader range of tobacco products and marketing in retail settings where they may seek products to aid in cigarette cessation (i.e., such broad product exposure could hinder cessation attempts); (3) consumer exposure to un-/under-regulated products (e.g., delta-8-THC, kratom); and (4) federal, state, and local regulations being undermined by consumer access to prohibited products online and via the mail. These concerns underscore the need for ongoing surveillance of how retailers and consumers respond to regulations.

## 1. E-Cigarette Use in the US

The global e-cigarette market is expected to grow from $15 billion in 2020 to $85 billion by 2028 [1]. The US represents nearly half the global e-cigarette market and is expected to increase from $7.4 billion in 2021 to $40 billion in 2028 [2], despite increasing e-cigarette sales and distribution restrictions [1]. These data underscore the prominence of e-cigarette retail and use in the US, representing the second most prevalent tobacco/nicotine product used among US adults [3].

While e-cigarettes may potentially reduce harm and aid in cigarette cessation [4,5], a significant concern is young adult e-cigarette use [3]. In those 18–24, current e-cigarette use increased from 5.2% in 2014 to 9.4% in 2020 (most recent data) [3]. E-cigarette use has also increased in those aged 25–30 [3,6,7,8]. Those entering young adulthood are also vulnerable; 11.3% of high school students reported past-month e-cigarette use in 2021 [9]. Young adults also demonstrate the highest prevalence of tobacco use, polytobacco use, other substance use [3,10,11,12,13], and polysubstance use (e.g., tobacco–cannabis co-use) [3,14,15,16,17,18,19]. Moreover, the aforementioned use prevalence rates are high among sexual/gender minority (SGM) groups (8.7 vs. 3.5% in heterosexuals) [3]. Despite low rates in racial/ethnic minorities (≤3.4 vs. 4.2% in Whites) [3], they may be at increased risk for dual use with other tobacco products [20].

## 2. E-Cigarette Retail & Marketing

The retail environment is a key driver of e-cigarette use [21,22,23,24,25,26,27]. Marketing aims to influence how and why consumers vape [28] (e.g., perceptions of safety or use for cessation of combustibles [29,30,31,32], social use, to achieve a “buzz”, for entertainment [31,32]) and to attract new users, promote continued use, and shape perceptions of products and their use [21,22,23,24,25]. This is particularly crucial for newer products, as the ways consumers are first exposed to a product is highly influential on short-term sales or gains [33,34].

How e-cigarettes are sold, and accessed by consumers, is a critical component of e-cigarette retail and marketing. E-cigarette devices are sold via various channels, including online, traditional retailers (e.g., convenience, food, liquor, drug stores), and tobacco specialty shops, particularly “vape shops” (that sell e-cigarettes but not other tobacco products [35]) and “smoke shops” (that sell nicotine vape and other tobacco products and accessories (e.g., hookah, cigars, pipes), and may sell cannabis-containing products [36,37]).

These retail settings are important for several reasons. First, retail settings, including both brick-and-mortar and online, are a prominent, effective source of tobacco advertising in youth [24,25,38,39,40,41] and adults [42,43,44]. Prior research showed that 68.6% of young adults reported past 30-day e-cigarette advertising/media exposure—most commonly seen on social media (43.4%) and at retailers (31.7%)—and e-cigarette advertising exposure was associated with e-cigarette use [45]. Moreover, research documents that prominent sources of young adults’ initial exposure to e-cigarettes are storefronts in one’s community or online, and that young adults most frequently purchased e-cigarette via vape shops and online [46]. Other US data suggest that young people were most commonly exposed to e-cigarette advertising at retailers (68%) and online (41%) [47]. In addition, after cigarettes, e-cigarettes are the most highly advertised tobacco product in stores [48]. These data underscore the need for surveillance of e-cigarette retailers, particularly specialty stores and online retailers, in order to comprehensively understand e-cigarette marketing and consumer exposure and perceptions, particularly as future regulations impact both.

## 3. E-Cigarette Regulation in the US

The past 6 years has marked major shifts in tobacco regulation in the US. In 2016, the Deeming Rule extended the FDA’s authority to e-cigarettes (and other tobacco products), including their manufacturing, retail, and marketing (e.g., prohibiting free samples, false/misleading ads) [49]. This set the foundation for FDA’s premarket tobacco product application and modified risk tobacco product application processes. The FDA issued the first marketing denial orders for some e-cigarette brands in August 2021, issued >323 marketing denial orders for >1,167,000 flavored e-cigarette products by September 2021, and first authorized an e-cigarette brand to continue sales in October 2021 [50].

A host of other federal, state, and local regulations that have impacted tobacco retail have also been advanced and/or gone into effect (see Table 1). One category of policies that impacted e-cigarettes in particular was that restricting e-cigarette sales and distribution. In January 2020, the FDA banned the sale of flavored cartridge-based e-cigarettes [51]; 5 states and 338 localities have implemented more comprehensive restrictions on flavors [52]. In December 2020, Congress extended the Prevent All Cigarette Trafficking (PACT) Act (which prohibits USPS delivery of cigarettes and smokeless tobacco to consumers) to include e-cigarettes [53]. However, prior research showed high rates of noncompliance with such regulation for cigarettes [54]. In 2020, federal legislation raised the minimum legal sales age for tobacco (including e-cigarettes) from 18 to 21 [55]. Yet, research has highlighted the following: high noncompliance rates in some states [56,57], underestimated noncompliance [57,58], gaps in FDA enforcement protocols [59,60,61], and particularly high noncompliance rates for e-cigarette sales [62], especially in tobacco specialty shops (such as smoke shops) [45,46,63] and online retailers [46,64], as well as for retailers in racially/ethnically diverse neighborhoods (i.e., Black, Latinx) [65]. Other federal legislation likely to continue being considered and potentially implemented includes: expanding flavored tobacco product restrictions [66], implementing pictorial health warning labels (in 2022) [67], and increasing federal tobacco taxes [68]. These initiatives could lead to changes in tobacco and e-cigarette user behaviors (e.g., quitting, product substitution), as well as changes at the point-of-sale (e.g., price-reducing promotions). State and local e-cigarette-related policies will also continue to evolve [69]. For example, given the increasing potency of e-liquids over time [70], some policy considerations might include placing limits on e-liquid nicotine concentrations [71]. However, such strategies are undermined by concerns that products, such as those labeled as zero nicotine, may not be accurately labeled [72].

## 4. Shifts in E-Cigarette Retail Settings

This commentary focuses on two changes in the e-cigarette retail setting that have cause for concern. One is the diversification of product offerings among vape shops, many of which now are more accurately termed smoke shops. The other is the increased prominence of online e-cigarette retail.

### 4.1. Vape Shops Shifting to Smoke Shops

In 2020, brick-and-mortar retailers (including vape and smoke shops) accounted for >80% of the e-cigarette market share globally [1]. Initially, vape shops were critical drivers of e-cigarette uptake, enabling users to sample products before purchasing [1], which was particularly relevant given the popularity of modular devices (that allow customization in nicotine levels, flavors, etc.) [1,2] and the increasing trend of users/retailers mixing their own e-liquid and/or circumventing flavor restrictions by combining flavored zero-nicotine liquid with nicotine concentrates [1].

Before the pandemic, and perhaps in response to FDA regulation and e-cigarette sales restrictions, vape shops were twice as likely as smoke shops to close from 2018 to 2019 [73]. In response, vape shops introduced other tobacco products and a variety of un- or under-regulated cannabis and delta-8 products [73,74,75].

For users who historically accessed e-cigarettes at vape shops, stores that transitioned to selling other tobacco represent a greater exposure to the marketing and availability of such products. Prior research indicates that the exposure to tobacco products, marketing, and retail undermines smoking cessation efforts [76,77]. Given that many people use e-cigarettes to try to quit smoking cigarettes or maintain abstinence from cigarettes, this is a major concern [31,78]. In fact, one study documented that among current e-cigarette users in the US surveyed in 2014 (*N* = 879) and 2016 (*N* = 743), a greater proportion of vape shop (40.2%) and internet customers (35.1%) versus retail (17.7%) and smoke shop customers (19.3%, *p* < 0.001) were former smokers [79]. Moreover, among those smoking a year ago, smoking cessation rates were higher for vape shop (22.2%) and internet customers (22.5%) than for retail customers (10.7%) and smoke shops (9.4%), and the proportion making a quit attempt was the lowest among smoke shop customers (50.6%) when compared to other groups, particularly vape shops (66.4%) [79].

Those going to smoke shops also are exposed to a greater variety of other products, in particular cannabis-based products. Policies and regulations enacted in the past decade have made many such products more accessible. For example, cannabidiol (CBD) products have increased in popularity and product diversity in recent years [80], since the Agriculture Improvement Act of 2018 [81] allowed for CBD derived from hemp (cannabis and cannabis derivatives with ≤0.3% delta-9-THC) to be legally purchased in most states. Further, 15 states and DC have legalized non-medical cannabis use for adults ≥21 [82]—this is notable here given increases in vaping cannabis [83] and vaping-related lung injury particularly connected to THC [84]. These products have been more prominent in smoke shops versus vape shops. For example, 6% of vape shops and 47% of smoke shops sold CBD and/or THC products in New Hampshire in 2017 [29]. Audits of vape shops in six US metropolitan areas indicated that 43% of vape shops sold CBD e-liquids and 23% sold other CBD products in 2018 [45]. Moreover, many vape shop owners show interest in the CBD/THC product sales [75] and increasingly brand their stores with cannabis-related terms (e.g., “haze”, “toke”) [73].

Compounding these concerns is that smoke shops provide access to unregulated cannabis-based products, such as delta-8-THC (see Figure 1) [85]. Delta-8-THC has a nearly identical chemical structure to delta-9-THC (the primary psychoactive agent in cannabis) and produces a similar “high” [86]. Most cannabis strains produce minimal amounts of delta-8-THC, but larger quantities can be synthesized by chemically converting CBD (including hemp-based CBD) to delta-8-THC, delta-9-THC, and other variants (see Figure 1) [86]. Evidence is limited regarding whether toxic or otherwise harmful substances are produced as byproducts in the conversion process, raising concerns about health implications [86]. This has led some jurisdictions, such as Colorado, to prohibit these conversion methods despite the legal recreational use of delta-9-THC [86,87]. In August 2020, the US Drug Enforcement Agency (DEA) declared that any processes that create delta-9-THC as a byproduct are in violation of federal law [88], although the extent to which DEA will act on this is unclear. This lack of clarity regarding the legal status of delta-8-THC has opened to door to opportunistic manufacturers [86], For example, while US states that permit recreational cannabis use require products to be sold by licensed dispensaries, many manufacturers sell bulk quantities of delta-8-THC and pre-mixed products that resemble cannabis products directly through their websites, and some also ship their products wholesale to retailers, particularly smoke shops [86]. Among the major concerns regarding these under-regulated product are their contents; one study documented that of 27 delta-8-THC products from 10 brands, none had accurate delta-8-THC labeling, which has significant implications for consumer safety [89].

Another under-regulated product frequently found in smoke shops is kratom (see Figure 1). Kratom is derived from the leaves of a tree native to Southeast Asia. In the US, kratom is typically sold in pill, powder, and liquid forms that can be easily dissolved or consumed with food [90] and is primarily marketed as an herbal medicine/supplement to treat a variety of ailments (pain, mental health, opioid withdrawal symptoms) as well as a “legal” or “natural” high and alternative to traditional opioids (e.g., “herbal speedball”) [90,91,92,93,94]. Kratom is sold through various brick-and-mortar retailers, particularly in smoke shops [90,91,92,93,94]. Although it has been available in the US for at least a decade, it has received greater public attention since the Centers for Disease Control and Prevention (CDC) reported a significant increase in kratom-related calls to poison control centers between 2011 and 2015 [95]. Among the 660 reported calls, 49 (7.4%) were classified as major, life-threatening events with some residual disability [95]. Moreover, given the mostly unregulated market for kratom products in Western countries, consumers may be exposed to adulterated or contaminated products, especially if purchased through websites or the darknet [96]. A number of countries have scheduled kratom because of its stimulant- and opioid-like effects and the established interaction of the alkaloid mitragynine with opioid receptors [96].

Delta-8-THC and kratom are just two categories of products that have emerged and been particularly prominent in smoke shops. Among others that have made a presence are “whippits”, Noz, and N2O—which contain nitrous oxide to be recreationally inhaled to achieve a high. These products are so readily available because some state laws (e.g., California) allow their sale to those older than 18, and/or allow sales intended for culinary and automotive use (see Figure 1). However, smoke shops have been selling these products for years [97]. These products represent just a few products represented at smoke shops—not seen at vape shops in general—and this list will likely grow.

Another major concern is youth access to products at smoke shops. An analysis of 2013–2014 PATH data indicated that, among youth, the most commonly reported source for e-cigarettes and hookah was smoke shops (49.8 and 51.3%, respectively) [98]. The most common ways of accessing other tobacco products were largely from social sources or convenience stores/gas stations [98]. Moreover, another study indicated that young adult e-cigarette users (ages 18–24) represented 3.2% of those who most frequently purchased products at vape shops, but a much larger proportion (12.9%) of those who most frequently purchased products at smoke shops [79]. One likely factor determining where youth access products via retailers is retailer compliance with age verification policies, as prior research indicates that, across 2016–2018, only ~25% of youth users who tried to buy tobacco products were refused sale because of age [99]. Although FDA regulation requires retailers to check ID for all persons ≤27 years old, one study found that 49.8% of tobacco and vape shops failed to check ID for underaged decoys, a higher rate of noncompliance than for other types of retailers [62]. While one study of FDA inspection data in four states from 2017 to 2019 indicated no differences between vape vs. smoke shops in age verification compliance [100], 2019 data from California indicated noncompliance rates of 13.4% in vape shops vs. 30.6% in smoke shops [57]. Moreover, surveillance of smoke shop promotional materials have identified fliers indicating “Back to School” specials that clearly target youth (see Figure 2).

### 4.2. Increasing Role of Online Retail for E-Cigarette Purchasing

Online e-cigarette retailers [1] are expected to be the fastest-growing segment in the future [2]. COVID-19 had a substantial impact on consumer behavior [101,102], with more consumers now purchasing goods online and using delivery services for groceries as well as alcohol and tobacco [1,2,103,104]. Specific to e-cigarette retailers, between 2019 and 2020, ~80% of vape shops reported a decline in revenue, with an average decline of 18% [104]. The greatest share of this decline was attributed to COVID-19 (with only ~7% attributed to state/local restrictions on flavored products) [104]. Prior research indicated that 53.2% of vape shops in six US metropolitan areas remained open during COVID-19 state-ordered business closures, with 31.4% offering pick-up/delivery services [103]. Notably, half of vape shops adapted to the pandemic by doing business online [104], which has been shown to be a high-priority across vape shops [105]. The online marketplace offers competitive pricing, convenience, and product variety, which is appealing to consumers and retailers [1,2]. Moreover, online retail can leverage the vast opportunities of marketing via the online environment, including via social media, which pose complexities and challenges to regulation and often target youth [106,107,108].

The online retail setting is concerning for multiple other reasons. Despite years of regulations restricting internet cigarette sales (per the PACT Act), enforcement of the Act is poor. US consumers can access tobacco products from all over the world via the Internet, undermining federal, state, and local policies [109]. Perhaps relatedly, reasons for consumers purchasing online, potentially from outside of their federal, state, or local boundaries, may be motivated by a desire to obtain tobacco products that are prohibited where they live. For example, research suggests that US residents, particularly those living in states with progressive restrictions on e-cigarettes (e.g., flavors), pursue (successfully) these products from online and international sources. Relatedly, online vendors frequently use language prohibited by the FDA, particularly language that implies reduced harm (e.g., “light”) [109]. In many cases, these vendors may be overseas, exposing buyers to widespread credit card fraud [109].

Another major concern is that poor vendor compliance and lack of shipper and federal enforcement leaves minors still able to obtain cigarettes and e-cigarettes online. In a 2014 study, minors received cigarettes from 32.4% of purchase attempts, all delivered by the US Postal Service (USPS) from overseas sellers. None failed due to age/ID verification; any failures were due to payment processing problems. USPS left 63.6% of delivered orders at the door; the remainder handed the product to minors with no age verification [109]. In another 2014 study that focused on e-cigarette purchases, minors successfully received deliveries of e-cigarettes from 76.5% of purchase attempts, with no attempts by delivery companies to verify their ages at delivery and 95% of delivered orders left at the door. All delivered packages came from shipping companies that, according to company policy or federal regulation, do not ship cigarettes to consumers. Of the total orders, only 5 of 80 youth purchase attempts were rejected based on age verification, resulting in a youth buy rate of 93.7% [109].

## 5. Conclusions

In conclusion, as regulatory and policy efforts at all levels (e.g., federal, state and local) to address e-cigarette use continue to evolve [49,51,53,66,68,69], the e-cigarette retail environment will continue to adapt by diversifying their products to include other tobacco, cannabis-based, and other products (e.g., kratom)—transforming into smoke shops—and by bolstering their online presence [73,74,75,105]. These important shifts in the tobacco specialty retail environment have concerning implications related to: (1) youth access; (2) consumer exposure to a broader range of tobacco products in retail settings where they may be seeking products to aid in cigarette smoking cessation (thus hindering cessation attempts); (3) consumer exposure to un- and under-regulated products (e.g., delta-8-THC, kratom); (4) US, state and local policies being undermined by consumer access to prohibited products online and via the US postal service. Thus, a rigorous examination of e-cigarette retail and marketing is needed to inform the development, implementation, and enforcement of such regulations. Critical aspects of surveillance efforts include: (a) assessing product availability, marketing, and price in brick-and-mortar and online retailers, as well as compliance with sales restrictions on flavored tobacco, and Tobacco 21; (b) monitoring how the tobacco industry responds to regulation and the downstream consequences for e-cigarette retailers (e.g., altering product offerings, promotional strategies, distribution methods) [74,75,105]; (c) identifying ways in which consumers, particularly youth and young adults, respond to changes in e-cigarette related policy and in e-cigarette retail and marketing (e.g., discontinuing use, switching to other tobacco products, changing e-cigarette products used or sources) [46,110,111]. In addition, ensuring that findings from such surveillance efforts have impact requires dissemination to policymakers, regulators, and public health practitioners, as well as to the public so that communities might be involved in addressing their own public health interests [112].

## Figures and Tables

**Figure 1 ijerph-19-08518-f001:**
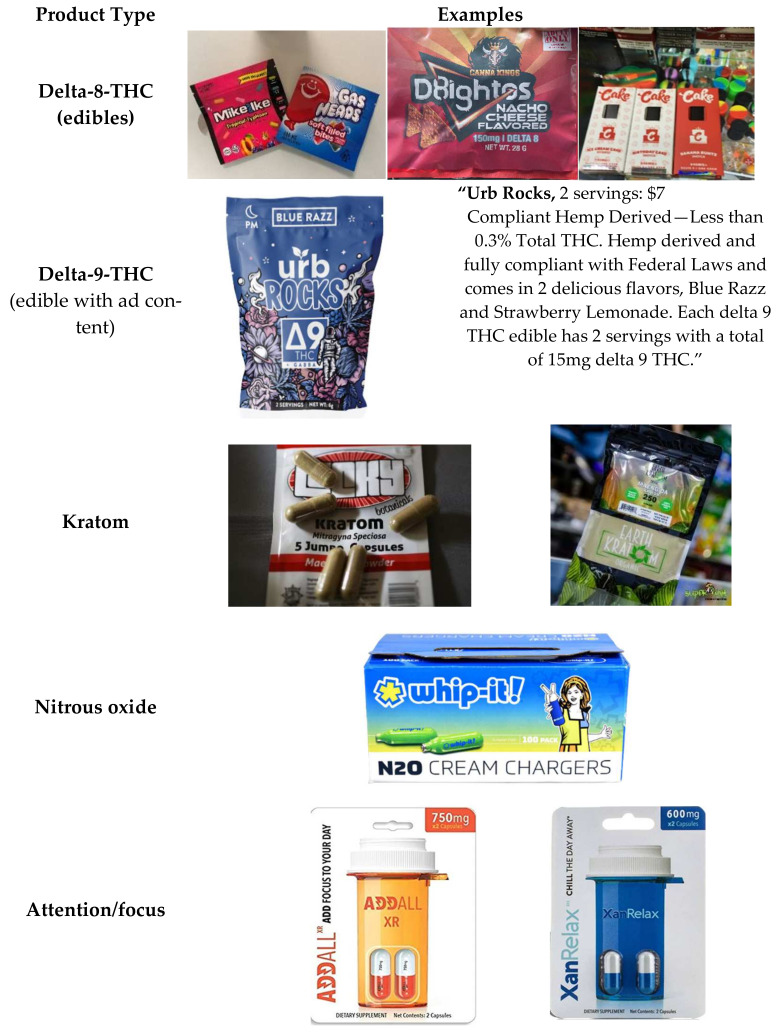
Products available at smoke shops.

**Figure 2 ijerph-19-08518-f002:**
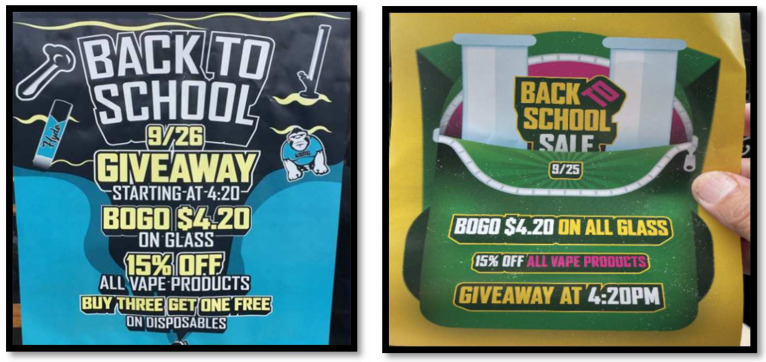
Fliers from smoke shops.

**Table 1 ijerph-19-08518-t001:** Recent and likely policy changes.

Federal:
2016—FDA Deeming Rule
2020—Flavored cartridge-based e-cigarette ban
2020—Minimum age increased to 21
2020—PACT Act extended to e-cigarettes
- Likely future—Tobacco/e-cigarette tax increase; Expanded flavored tobacco product sales restrictions; Pictorial health warnings; etc.
**State/local (vary across settings):**
- Restrictions on sales of: flavored e-cigarettes, flavored tobacco, or all e-cigarettes (e.g., California)
- Tax rates on e-cigarettes vs. other tobacco
- Required tobacco/e-cigarette retailer licensure
- Inclusion of e-cigarettes in state definitions of tobacco and in tobacco-/smoke-free policies

## Data Availability

Not applicable.

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
