# Peer review of "The Reshaping of the E-Cigarette Retail Environment: Its Evolution and Public Health Concerns"

_ijerph, 2022, doi:10.3390/ijerph19148518_

Round 1

Reviewer 1 Report

Major Comments: This is a very nice review of the retail factors influencing the purchase of e-cigarettes and other vaping and psychotropic agents.  The focus is appropriately on the purchase of these products by youth and the regulations (federal, local, and state) that govern but don’t completely restrict the illegal purchase of these products by those under 21 years of age.  The only major concern is the focus on U.S. regulation and use of e-cigarettes and other products.  Although a more global view would be valuable, it is understandable that the focus is on one country given the rapidly changing regulatory environment, thus making a ‘global review’ extremely difficult.

Minor Comments:

Line 17 – cessation of what?

Line 29 – This is a bit unclear to the reader: “to aid in cigarette cessation (thus hindering cessation attempts” First aid in cessation and then hindering?

Line 31 – mail is unclear – add USPS?

Line 48 – Unclear: “high rates” of what products?

Lines 53-55 – Awkward sentence that says marketing aims to expand markets.

Line 59 – delete ‘the’

Line 79 – Global issues were brought up before, so perhaps add ‘in the U.S.’ to this title as everything following is U.S.-centric.

Line 80 – update ‘5’ years if necessary.

Line 102 – ‘advance’ is unclear

Table 1 – Why are there ‘-‘ before each year?  Also, there are some extra spaces before e-cigarettes in a few places.

Lines 125-6 and 131-2: Duplicate sentences so delete one.

Line 147 – update 15 states if needed.

Line 157 – This reviewer is not a chemist but it seems that saying the psychotropic compound in THC is incorrect.  Isn’t delta9-THC the compound?

Line 198 – ‘allows’ and ‘intended’ are not grammatically correct.

Author Response

We would like to thank the reviewers for their thoughtful feedback. Below we respond to each point. We appreciate the opportunity to enhance the quality of this manuscript by doing so.

Reviewer 1

  1. Major Comments: This is a very nice review of the retail factors influencing the purchase of e-cigarettes and other vaping and psychotropic agents.  The focus is appropriately on the purchase of these products by youth and the regulations (federal, local, and state) that govern but don’t completely restrict the illegal purchase of these products by those under 21 years of age.  The only major concern is the focus on U.S. regulation and use of e-cigarettes and other products.  Although a more global view would be valuable, it is understandable that the focus is on one country given the rapidly changing regulatory environment, thus making a ‘global review’ extremely difficult.

Response: Thank you sincerely for noting this as a concern, but also as a very complex issue. Even within the US, this is extremely complex, given the vast differences in regulations across states. For this reason, we maintained a domestic focus. Again, we appreciate this reviewer acknowledging this.

Minor Comments:

  1. Line 17 – cessation of what?

Response: We have specified “cigarette” cessation.

  1. Line 29 – This is a bit unclear to the reader: “to aid in cigarette cessation (thus hindering cessation attempts” First aid in cessation and then hindering?

Response: We have revised to clarify: “consumer exposure to a broader range of tobacco products and marketing in retail settings where they may seek products to aid in cigarette cessation (i.e., such broad product exposure could hinder cessation attempts)”.

  1. Line 31 – mail is unclear – add USPS?

Response: We are referring to the broad range of carriers – including domestic and international, which may include USPS. We chose not to specify further in the abstract, but this is expanded upon in the manuscript.

  1. Line 48 – Unclear: “high rates” of what products?

Response: We have clarified: “Young adults also demonstrate the highest prevalence of tobacco use, polytobacco use, other substance use,3,10-13 and polysubstance use (e.g., tobacco-cannabis co-use).3,14-19 Moreover, the aforementioned use prevalence rates are high among sexual/gender minority (SGM) groups (8.7% vs. 3.5% in heterosexuals).3

  1. Lines 53-55 – Awkward sentence that says marketing aims to expand markets.

Response: Thank you for noting this. We have revised as follows: “Marketing aims to influence how and why consumers vape28 (e.g., perceptions of safety or use for cessation of combustibles,29-32 social use, to achieve a “buzz”, for entertainment31,32) and to attract new users, promote continued use, and shape perceptions of products and their use.21-25

  1. Line 59 – delete ‘the’

Response: We have done so.

  1. Line 79 – Global issues were brought up before, so perhaps add ‘in the U.S.’ to this title as everything following is U.S.-centric.

Response: Thank you for this suggestion. We have done so.

  1. Line 80 – update ‘5’ years if necessary.

Response: Thank you – we realize that this is now 6 years and have edited accordingly.

  1. Line 102 – ‘advance’ is unclear

Response: We have specified: “Other federal legislation likely to continue being considered and potentially implemented includes….”

  1. Table 1 – Why are there ‘-‘ before each year?  Also, there are some extra spaces before e-cigarettes in a few places.

Response: Thank you for noting this. We have removed the dashes before the years and removed the extra spaces.

  1. Lines 125-6 and 131-2: Duplicate sentences so delete one.

Response: Thank you for noting this. We have addressed this duplication.

  1. Line 147 – update 15 states if needed.

Response: Thank you for noting this.

  1. Line 157 – This reviewer is not a chemist but it seems that saying the psychotropic compound in THC is incorrect.  Isn’t delta9-THC the compound?

Response: Thank you for noting this; we have revised to say: “Delta-8-THC has a nearly identical chemical structure to delta-9-THC (the primary psychoactive agent in cannabis) and produces a similar “high”.”

  1. Line 198 – ‘allows’ and ‘intended’ are not grammatically correct.

Response: We appreciate this note. We have revised to say: “These products are so readily available because some state laws (e.g., California) allow their sale to those older than 18, and/or allow sales intended for culinary and automotive use.”

Reviewer 2 Report

Given the confusion surrounding electronic cigarette and cannabis legislations, this commentary is very timely and important. The commentary written by Berg et al is extremely well-written and easy to follow.

The authors do an excellent job of supplying information without coming across as biased or one-sided.

Although it is extremely thorough, this reviewer requests that the authors touch on the changing levels of nicotine potency in e-cigarette liquids, as well as the assumption that a "0% nicotine" e-liquid really contains 0% nicotine.

These factors have certainly also played into the reshaping of the e-cigarette retail environment, consumption patterns of young individuals, and other factors mentioned in this commentary. 

Author Response

We would like to thank the reviewers for their thoughtful feedback. Below we respond to each point. We appreciate the opportunity to enhance the quality of this manuscript by doing so.

Reviewer 2

  1. Given the confusion surrounding electronic cigarette and cannabis legislations, this commentary is very timely and important. The commentary written by Berg et al is extremely well-written and easy to follow. The authors do an excellent job of supplying information without coming across as biased or one-sided.

Response: We sincerely appreciate this comment.

  1. Although it is extremely thorough, this reviewer requests that the authors touch on the changing levels of nicotine potency in e-cigarette liquids, as well as the assumption that a "0% nicotine" e-liquid really contains 0% nicotine. These factors have certainly also played into the reshaping of the e-cigarette retail environment, consumption patterns of young individuals, and other factors mentioned in this commentary. 

Response: Thank you for this suggestion. We have written: “State and local e-cigarette related policies will also continue to evolve.69 For example, given the increasing nicotine concentration levels in e-liquids over time,70 some policy considerations might include placing limits on e-liquid nicotine concentrations.71 However, such strategies are undermined by concerns that products, such as those labeled as zero nicotine, may not be accurately labeled.72